# Recreational Drugs and the Risk of Hepatocellular Carcinoma

**DOI:** 10.3390/cancers14215395

**Published:** 2022-11-02

**Authors:** José M. Pinazo-Bandera, Miren García-Cortés, Antonio Segovia-Zafra, María Isabel Lucena, Raúl J. Andrade

**Affiliations:** 1Service of Gastroenterology and Hepatology, University Hospital Virgen de la Victoria, Universidad de Málaga, 29010 Málaga, Spain; 2Instituto de Investigación Biomédica de Málaga y Plataforma en Nanomedicina-IBIMA Plataforma Bionand, 29010 Málaga, Spain; 3Centro de Investigación Biomédico en Red Enfermedades Hepáticas y Digestivas (CIBERehd), 29010 Madrid, Spain; 4Service of Clinical Pharmacology, University Hospital Virgen de la Victoria, Universidad de Málaga, 29010 Málaga, Spain

**Keywords:** recreational drugs, illicit drugs, herbals, hepatocellular carcinoma

## Abstract

**Simple Summary:**

Hepatocellular carcinoma represents an important contributor to the global cancer-related burden, and liver cirrhosis is the main risk factor for its development. Conventional or illegal drug consumption is a potential but infrequent cause of cirrhosis. However, the causal relationship between recreational drugs and the risk of developing liver cancer has not been studied in detail thus far. The aim of this review is to synthesize the available published evidence on legal and illegal recreational drug use and the risk of hepatocellular carcinoma and other liver tumors. Expanding our knowledge about the contributions of these substances to the appearance of liver cancers is important for combatting this preventable cause of cancer.

**Abstract:**

Recreational or aesthetic drug use is a distinctive behavior of humans, principally attested in the last century. It is known that recreational and illegal drugs are major contributors to the universal morbidity rate worldwide. Many of these substances have a well-established hepatotoxic potential, causing acute or chronic liver injury, liver fibrosis and cirrhosis, but their implications for hepatocellular carcinoma or other varieties of liver tumors are little known. In this article, we perform an extensive literature review, aiming to provide updated information about recreational drug use and the risk of developing liver tumors. Khat use and pyrrolizidine alkaloid consumption (present in some natural plants) have been linked to liver cirrhosis. Kava intake is associated with different liver tumors in animal models but not in humans. Cannabis’ potential to accelerate liver fibrosis in chronic hepatitis is controversial according to the existing data. Cigarette smoking is an important contributor to hepatocellular carcinoma, and anabolic androgen steroids are well-defined causes of a variety of liver cancers and other hepatic tumors. Long-term follow-up studies of subjects who have developed injuries in association with the use of recreational drugs are warranted so as to better define the risk of developing hepatocellular carcinoma in association with these substances and, thus, to implement health care policies to combat this preventable cause of cancer.

## 1. Introduction

Hepatocellular cancer (HCC) is the sixth most frequently diagnosed cancer and the third cause of cancer-related death worldwide, according to the recent registries. In 2020, approximately 724,800 new cases of HCC were diagnosed, with a total of 664,000 deaths in 185 countries from all over the world [1]. Therefore, HCC represents a sizeable contributor to the global cancer-related burden [2]. The incidence of HCC increases substantially in advanced aging population, with a peak in seventh decade of life, except for Chinese and black African patients, whose mean ages at diagnosis are considerably low [3,4]. Regarding the sex preponderance, there is a ratio of around 2.5:1 (male:female) [1]. Only 10% of HCCs occur in patients without any known etiology of liver disease, whilst approximately 90% are associated mainly with chronic viral hepatitis, alcohol abuse, aflatoxin exposure and metabolic-associated fatty liver disease (MAFLD) [5]. Nowadays, hepatitis B (HBV) infection has become the main cause of HCC, accounting for roughly 54% of cases globally. Nevertheless, the incidence and prevalence of MAFLD-related HCC are expected to rise dramatically due the growing obesity pandemic [5,6]. The estimated annual incidence of HCC among patients with non-alcoholic-steatohepatitis-related cirrhosis is 0.5–2.6%. Thus, MAFLD may soon become the foremost global cause of HCC [6].

Cirrhosis is the most important risk factor for the development of HCC [7]. Thus, it is estimated that one out of three patients with cirrhosis will suffer from HCC during their lifetime [7]. Although chronic viral hepatitis, chronic alcohol intake and MAFLD are responsible for most cases of cirrhosis, any other cause of cirrhosis (e.g., drugs or herbal causes) might result in hepatocellular carcinoma [4]. It is well-known that chronic liver damage caused by conventional drugs can lead to severe liver fibrosis and even liver cirrhosis. Once this state has been reached, it is not unlikely to develop into HCC due to the patient’s natural history of cirrhosis [8]. However, neither in the case of conventional medications nor the illicit use of recreational drug or herbs is there robust evidence regarding their associations with the risk of HCC.

Illicit drug and tobacco use, together with alcohol intake, are major contributors to the universal morbidity rate worldwide [9]. Despite the well-known risks of systemic diseases (such as malignancies, lung and cardiovascular disorders, etc.), tobacco consumption remains a serious public health concern [10]. According to World Health Organization (WHO), cigarette smoking kills more of the European population than any other preventable cause [10]. Likewise, substance use disorders, along with mental illnesses, are the first cause of the health burden in young people, accounting for nearly 20% of all disability-adjusted life years. Furthermore, illicit substance consumption carries the risk of significant physical and social disturbances, such us traffic accidents, violent behaviors, and increased suicide rates, among others [11,12]. Thus, among injection drug addicts, the proportion of chronic hepatitis cases is not negligible, causing an important burden of liver cirrhosis and, consequently, of virus-related hepatocellular carcinoma [13]. 

On the other hand, in modern society, physical appearance is increasingly acquiring an overestimated value, favoring the popularity of illegal and illicit substances used to strengthen and enlarge the muscles for aesthetic purposes [14]. Anabolic androgenic steroids (AAS), nowadays, have very limited medical indications [15]. However, AAS are typically used by 20–40-year-old people, predominantly males, who illicitly consume these drugs for recreational purposes in the context of gym practice or weight training [16,17].

Information about the use of several recreational substances and the development of acute and chronic liver diseases has become available in recent decades. However, except for alcohol-related HCC, which is beyond the scope of this review, there is scarce evidence on the relationship between the exposure to these agents and the risk of HCC.

The aim of this review is to provide updated information and to combine the available published evidence on legal and illegal recreational drug use and the possibility of developing HCC (see Table 1 and Figure 1). For this purpose, we performed an extensive literature search of MEDLINE-PUBMED with the terms ‘hepatocellular carcinoma’, ‘hepatocellular adenoma’, ‘liver tumor’ and ‘drug-induced liver injury’, ‘herbal’, ‘herbs’, ‘dietary supplements’, ‘tobacco’, ‘cannabis’, ‘cocaine’, ‘heroin’, ‘methamphetamine’ and ‘anabolic androgenic steroids’.

## 2. Conventional Drugs and Hepatocellular Carcinoma (HCC)

Drug-induced liver injury (DILI) is a relatively infrequent liver disorder but can be severe and, in some cases, fatal (resulting in liver transplant or death). The most common phenotype of DILI is a viral-like hepatitis syndrome. Nevertheless, this entity can mimic any other acute and chronic liver disorder [84]. 

Chronic DILI, which may result in fibrosis and cirrhosis in some instances, is a well-known consequence of toxic liver damage [8]. In the prospective Spanish DILI registry, 25 out of 298 (8%) patients followed for at least 1 year developed chronic DILI. Liver biopsy was available for 16 patients, and 7 showed histological findings compatible with cirrhosis. In this study, the drugs causing cirrhosis were atorvastatin, bentazepam, ebrotidine, clopidogrel/atorvastatin, amoxicillin-clavulanate/ibuprofen and ranitidine [18]. A few drugs have shown a potential to induce progressive liver injury resulting in cirrhosis, including ebrotidine, amiodarone, methotrexate and nitrofurantoin. Ebrotidine is an H2 receptor antagonist with gastroprotective activity, which was removed from the market due to instances of serious liver injury, including fulminant liver failure and cirrhosis [19]. Amiodarone-related cirrhosis usually occurs in patients on long-term treatment with this agent [20]. Additionally, methotrexate-induced liver injury is a rare cause of severe liver fibrosis, with some disparity between the prior published series [21]. Decompensated liver cirrhosis caused by this agent seems to be extremely infrequent, occurring in the setting of an MAFLD-like phenotype [22]. On the other hand, nitrofurantoin is a drug with a well-described hepatotoxic potential. Its spectrum of liver damage is very broad, including an autoimmune-like DILI phenotype and, occasionally, severe fibrosis/cirrhosis, also associated with long-term therapy [23,24]. Although the capacity of these drugs to cause cirrhosis is well-documented, the risk of HCC-related cirrhosis due to conventional medications is still unknown because of the rarity of this phenotype and the limited follow-up of the cases reported [8,24]. Nonetheless, patients with drug-related cirrhosis, as in the case of subjects with cirrhosis of other etiologies, should probably undergo HCC surveillance.

Paradoxically, amiodarone has recently been identified as a new autophagy inducer, suppressing HCC growth in vitro [25]. In fact, patients with HCC and cardiac arrhythmia on amiodarone therapy seem to have a lower risk of HCC-related mortality than non-amiodarone users [26].

## 3. Herbal and HCC

### 3.1. Khat and HCC

Khat (*Catha edulis*) is a plant with psychoactive effects similar to those of amphetamine. Indeed, it is popularly known as the “natural amphetamine”. Specifically, this herb is one of the most widely consumed herbs in the whole world, with users usually chewing its leaves [85]. The mean age of khat users is approximately the second decade of life, with a significant predominance of the male gender [86]. Khat is a legal and socially accepted substance used recreationally in East African and Middle Eastern countries (mainly Somalia, Ethiopia and Yemen) [86]. It is estimated that it has a daily consumption by more than 20 million people [87]. However, Khat is considered a drug of abuse by the WHO, and its sale is forbidden in almost all Western countries, including the United States of America (USA) and European Union [86].

Khat is a well-established hepatotoxic agent, with case reports and case series implicating the herb in acute and chronic liver injury, and it is defined as category B in terms of its hepatotoxicity potential in LiverTox (this xenobiotic has been reported and is known or highly likely to cause idiosyncratic liver damage, with 12–50 previously published cases) [27,88]. Although it is not clear whether this toxic potential is due to its constituents or a direct effect of the pesticides/preservatives used for cultivating and transporting the herb [28]. The mechanism of khat-related liver injury remains to be elucidated. Animal model studies have shown that khat can cause acute hepatitis, and that chronic active hepatitis and fibrotic liver disease are linked to long-term khat exposure in rats [89]. On the other hand, vasoconstriction due to cathinone, one of its components, has been suggested as a plausible mechanism of the liver injury [90]. It is worth mentioning that, in humans, the pattern of liver damage associated with khat typically has autoimmune features, with frequent autoantibody presentation and, histologically, chronic hepatitis and fibrosis [29,30,31,32,91]. Indeed, a cross-sectional study carried out in Ethiopia—one of the countries with the highest prevalence of khat intake—showed, interestingly, an eye-catching ratio of cirrhosis of apparently unexplained causes (55%). The authors noted that the widespread consumption of khat in this area, together with histological features of toxic injury in the liver biopsy performed among a subgroup of patients without an established etiology, suggest a probable causal association [33]. Similar data have been published in other regions with a high prevalence of khat use, such as Somaliland [34]. Nonetheless, a careful review of all the published reports on khat-induced liver injury does not provide sufficient evidence to link khat consumption with the development of HCC [27,28,29,30,31,32,33,34,35,36,37,38].

### 3.2. Kava and HCC

Kava (or kava-kava) is a herb extracted from the roots of the plant named *Piper methysticum.* Kava has been taken as a recreational beverage in Oceania for centuries, and more recently, it has been used in its concentrated form or as an infusion to alleviate anxiety disorders [92,93].

The hepatotoxicity of kava has been recognized in recent decades, with an important fraction of sever and fatal cases, and it is thus included in category A (highest hepatotoxic potential) in LiverTox (the xenobiotic is well-known, with more than 50 cases published in the literature) [39,94]. Several experimental studies on in vivo mouse models have shown the liver oncogenic potential of kava. In 2011, Behl et al. found an increase in the incidence of hepatoblastoma (dose-dependent) in male mice and an increment in hepatocellular carcinoma and adenoma detection (non-dose-related) in female mice. Moreover, non-neoplastic lesions (such as hepatocellular hypertrophy) were detected in the livers of mice, and such findings were confirmed in an independent study, which also identified a considerable rise in the rates of liver cancer and non-malignant liver lesions in mice of both sexes after kava exposure [40,41]. In contrast, in humans, there is no convincing published evidence linking kava intake to HCC thus far [39,42,43,44,45,46,47,48,49,50,51,52,53]. This apparent discrepancy could be related to the low quality of the design of these studies (observational, lacking long-term follow-up) that precluded the drawing of firm conclusions.

### 3.3. Other Herbs and HCC

Other herbs have been associated with chronic liver damage, even causing liver fibrosis and cirrhosis. In a recent review of the published case reports on herbal-induced liver injury in Latin America, a case of chronic hepatitis with a cholestatic and granulomatous pattern and cirrhotic processing due to *Centella asiatica* was identified [54,95]. *Centella asiatica* (or *Gotu kola*) is a traditional Chinese plant frequently used in Southeast Asia that occasionally is taken to facilitate weight loss, among other applications [96]. In the same systematic review, another case of herb-induced hepatotoxicity with a chronic course due to *Crotalaria juncea* was also included [54]. This herb is traditionally used as household remedy for several medical conditions, and it is known to cause sinusoidal obstruction syndrome [97]. *Crotalaria juncea* contains pyrrolizidine alkaloids among its components [54], which have been clearly related to veno-occlusive disease and, hence, to cirrhosis [54,55]. Nevertheless, there is no published evidence to associate these herbs with the risk of HCC.

## 4. Cigarette Smoking and HCC

Tobacco is a preventable leading risk factor for preterm morbidity and mortality, with sturdy evidence based on widespread population-based studies worldwide [2]. Nowadays, hepatologists often focus on recommending smoking discontinuation for patients with fatty liver disease, due to the frequent associated cardiovascular comorbidities, and for liver transplant recipients, because of its strong relation with cancer [56].

Several chemicals, such as 4-aminobiphenyl, polycyclic aromatic hydrocarbons and nitrosamines, among others, that are present in cigarettes are generally metabolized via CYP2E1, leading to carcinogenic substances [98,99,100]. This could support, per se, the oncogenic potential of cigarette smoking, but in addition the significant synergistic effect of tobacco–obesity, the effects of tobacco–chronic hepatitis and tobacco–alcohol intake are also commonly seen [101,102]. However, the association between smoking and MAFLD-related HCC has not been thoroughly studied [56].

Several observational case–control and cohort studies have reported an increased incidence of HCC in smoker populations, stratified by the severity of smoking [56]. This association was subsequently confirmed by three meta-analyses. In 2009, the first meta-analysis, including 38 cohort studies and 58 case–control studies, showed an adjusted relative risk of HCC of 1.12 (95% CI 0.78–1.60) for former smokers and 1.51 (95% CI 1.37–1.67) for current smokers [57]. In 2017, a more powerful meta-analysis including 81 observational studies reported an odds ratio of HCC of 1.39 (95% CI 1.26–1.52) for former smokers and 1.55 (95% CI 1.46–1.65) for current smokers [58]. Lastly, in 2018, the Liver Cancer Pooling Project consortium (from USA) updated information on this issue, whose strength was the prospective design of the 14 studies included. This study reported a hazard ratio of HCC development of 1.24 (95% CI 1.08–1.43) for former smokers and 1.86 (95% CI 1.57–2.20) for current smokers [59]. In summary, tobacco smoking is a strong risk factor for HCC, either alone or in association with other carcinogenic agents, such as alcohol, and it probably potentiates other pre-cancer settings, such as MAFLD. 

## 5. Cannabis and HCC

Approximately 200 million people consume cannabis, being the most widely used illicit substance worldwide [103]. Although marijuana consumption has not been recognized as a direct risk factor for the development of HCC thus far, its role in the progression of liver fibrosis in chronic liver disease is controversial. In 2005, Hézode et al., in a retrospective analysis, concluded that daily cannabis smoking was related to higher risk of the progression of fibrosis in patients with chronic hepatitis C [60]. On the contrary, two prospective studies (with 690 and 575 patients who had hepatitis C virus (HCV)–human immunodeficiency virus coinfection, respectively) concluded that cannabis use did not accelerate the progression of liver fibrosis in these cohorts [61,62]. Finally, a meta-analysis carried out in 2019 (which included two observational prospective and seven retrospective cross-sectional studies) concluded that marijuana use was not associated with a more accelerated progression of liver fibrosis [63]. 

## 6. Other Illicit Drugs and HCC

There is considerable evidence regarding the contributions of injection drug use to the global mortality burden of liver cancer, which has come to light in the last two decades, especially in Latin America, Eastern Europe and Central Asia [104]. However, this is probably the result of the increased transmission of HBV and, above all, HCV in this specific population, associated with an injected route of administration, rather than the intrinsic oncogenic potential of different types of consumed drugs [104].

Cocaine is a well-defined cause of acute liver injury. In the mechanism of liver injury, the conversion to a toxic metabolite during the process of the cytochrome P450 metabolism probably plays an important role [105]. The typical phenotype of cocaine-related liver injury is acute hepatic necrosis, and the histological changes resemble ischemic hepatitis or liver injury caused by hyperthermia, factors involved in the hepatotoxic effects of cocaine-related liver impairment [106]. There are no published data thus far associating cocaine consumption with a higher risk of HCC.

Regarding heroine and heroin derivatives, after multiple decades of significant worldwide use of these substances, they have not convincingly been shown to be directly responsible for liver injury. There are only preliminary data on animal models regarding an elevation in the aspartate-aminotransferase (AST) and alanine-aminotransferase (ALT) levels and histopathological liver injury [107]. In humans, alterations in the liver enzymes in heroin users have been attributed to other toxic habits, such as alcoholism, as well as coexisting chronic HVB or HCV [108]. Likewise, heroin use has not been linked to an increased risk of HCC. 

On the other hand, amphetamines are illicit drugs with significant consumption rates in the present time. These substances and their derivatives are central nervous system stimulants, often used as “social” drugs in party settings. Methylenedioxymethamphetamine is one of the most widely consumed. The abuse of these substances has been associated with severe liver injury, commonly accompanied by hyperthermia, respiratory failure and shock [109]. This compound is a well-defined cause of liver injury, belonging to category A in Livertox [110]. The histological pattern of the damage mimics that of the liver damage related to acetaminophen overdose [110]. Nonetheless, amphetamines use has not been related to a higher risk of HCC.

## 7. Anabolic Androgenic Steroids and HCC

Anabolic androgenic steroid consumption is typically linked to bodybuilding, and these drugs are often offered in private gym contexts and sold as dietary supplements [111]. Although the majority of AAS users are male, its use among women is increasing. The lifetime prevalence of AAS intake has been estimated to be around 0.1% in females [112]. AAS use has been related to an array of adverse effects, including cardiac impairment (sudden death and major cardiovascular risk), endocrine disease (hypogonadism after abrupt withdrawal and sexual dysfunction), neuropsychiatric effects (depression and AAS dependence syndromes), nephrotoxicity and liver damage (acute and chronic) [16,113].

Anabolic steroids have a well-known hepatotoxic potential, which has been reported extensively in the literature. The proportion of AAS cases in the DILIN registry (from the USA) is 5% of the total cases [114]. In the Spanish and the Latin American DILI cohorts, the percentage of AAS-related liver damage is 2.3% and 5%, respectively [115,116]. Liver damage related to anabolic steroids is typically cholestatic, with marked and prolonged jaundice (usually more than 3 months). Hy’s law, ALT ≥ 5x upper limit of normal + Total bilirubin ≥ 2 mg/dL, is commonly fulfilled in hepatocellular cases, but fulminant liver failure has not been reported, and gradual recovery is the rule. Acute kidney injury accompanies AAS-DILI in 14%–24% of cases, and it is related to the high bilirubin levels. Therefore, its resolution falls in parallel with bilirubin normalization [116,117,118]. Furthermore, the long-term use of anabolic steroids has been linked to peliosis hepatis, a rare vascular syndrome in which enlarged sinusoids and cysts in the hepatic parenchyma are observed [119]. 

Anabolic steroids are associated with hepatocellular adenoma and/or carcinoma, with a total of 29 cases of AAS-related liver adenoma/carcinoma reported in the literature [64,65,66,67,68,69,70,71,72,73,74,75,76,77,78,79,80,81,82,83]. Subjects who developed liver tumors were more frequently males (71%) and the predominant ethnical group was Caucasian (67%) in the few cases where this information was reported. The mean age of patients was 29 years (4–69). A total of 16 patients took AAS in monotherapy, 10 patients used at least more than one anabolic steroid, and three cases did not provide such information. The specific type of anabolic androgenic steroids observed more frequently included oxymetholone (10 cases) followed by different forms of testosterone (9 cases), nandrolone (6 cases) and stanozolol (5 cases). The main route of administration was oral in 17 patients, and the mean treatment duration (months) was 82.5 (6–240) (see Table 2 and Appendix A) [64,65,66,67,68,69,70,71,72,73,74,75,76,77,78,79,80,81,82,83].

Although the main application of AAS use was recreational consumption in the bodybuilding context (n = 9), medical applications followed, including Fanconi´s anemia (n = 5), hypoplastic-aplastic anemia (n = 5) and hereditary angioedema (n = 3) (see Table 2). From 1961 to 1999, AAS inducing HCC were strictly prescribed for medical conditions (mainly hematological diseases). In the 2000s, the consumption of AAS exceeded its therapeutic use, and the main reasons for its illicit use were bodybuilding and the desire to gain muscle mass (75% since 2005) [64,65,66,67,68,69,70,71,72,73,74,75,76,77,78,79,80,81,82,83].

Five patients (17%) had evidence of underlying liver disease. Most cases had hepatocellular carcinoma (n = 16; 55%), and 11 cases had liver adenoma (38%) (see Table 2). Typically, patients presented with multiple liver-space-occupying lesions (n = 24; 83%), and clinical manifestations developed in 26 cases (90%), mainly abdominal pain (59%) and hepatomegaly (55%), with jaundice being unusual (18%) [64,65,66,67,68,69,70,71,72,73,74,75,76,77,78,79,80,81,82,83].

With regard to biochemical parameters, the mean AST level was 236 IU/L (34–1437), the mean ALT level was 579 IU/L (71–2457), the mean alkaline phosphatase (ALP) level was 608.37 IU/L (55–3402), the mean Ὑ-glutamyl transferase (GGT) level was 192.75 IU/L (60–463) and the mean total bilirubin level was 3.49 mg/dl (0.3–18.8). Only three patients had increased levels of α-fetoprotein. Thus, a normal α-fetoprotein level is more common in AAS-related liver tumors (55.17%) compared to other prior published series of non-AAS-related HCC cases, in which the proportion of patients with high levels of α-fetoprotein (at least >10 ng/mL) reached 83% of the total [120]. Histological data are available in Table 2 [64,65,66,67,68,69,70,71,72,73,74,75,76,77,78,79,80,81,82,83].

The standard presentation of AAS-related HCC is a non-metastatic tumor at diagnosis (only one case of bone metastasis). Overall, survival data are not available for most of these patients. Anabolic steroids were discontinued in 55% of cases, and oncological or surgical therapies were relatively heterogeneous and closely related to the decision of the Tumor Committee of each single center (see Appendix A) [64,65,66,67,68,69,70,71,72,73,74,75,76,77,78,79,80,81,82,83].

To summarize, HCC associated with AAS use is typically a consequence of long-term exposure to these substances, with a symptomatic debut (abdominal pain and hepatomegaly) in relatively young patients with normal α-fetoprotein levels and multiple liver tumors (several liver-space-occupying lesions), with no metastatic disease.

It should be emphasized that around 4% of all hepatocellular adenomas finally become hepatocellular carcinomas [81]. General exome sequencing has identified recurrent somatic activating mutations in some genes (FRK, JAK1, gp130 and β-catenin), which may be responsible for the adenoma–carcinoma transformation. In addition, an integrative analysis of the liver adenoma–HCC transformation revealed that β-catenin mutation is a very early alteration, whereas TERT promoter mutations are associated with the last stage of the adenoma–carcinoma sequence [121]. It is unclear whether AAS can induce de novo HCC instead of the adenoma–carcinoma sequence pathway. The positive testosterone receptor has been detected in tumor biopsy, which may support the adenoma–carcinoma escape theory [122].

## 8. Conclusions and Future Prospects

This is the first manuscript in the literature that attempts to synchronize the evidence on recreational drug use and the risk of HCC. Recreational drugs are increasingly used, and although they are seldom associated with the development of HCC, the carcinogenetic risks for the liver posed by many of these substances are currently unknown. Whereas cigarette smoking and anabolic androgen steroids are well-established causes of HCC, the liver oncogenic risk of khat, kava and marijuana consumption has not been yet established. Long-term follow-up studies on subjects who have developed injuries in association with the use of recreational drugs are warranted in order to better define the risk of developing HCC due to these substances. Nonetheless, HCC related to exposure to recreational drugs is fully preventable if appropriate socio-political, legislative and sanitary strategies are implemented.

## Figures and Tables

**Figure 1 cancers-14-05395-f001:**
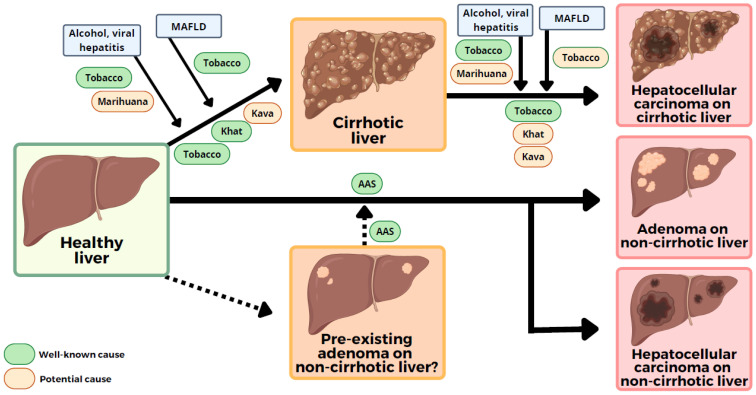
Well-defined and *potential* pathways for the development of recreational-drug-related liver tumors. This figure shows the potential role of recreational drugs in the development of liver tumor or their interactions in the cirrhosis–hepatoma sequence. Green: well-known cause of liver tumor. Orange: potential cause of liver tumor. Abbreviations: AAS, anabolic androgenic steroid; MAFLD, metabolic-associated fatty liver disease. Created with BioRender.com.

**Table 1 cancers-14-05395-t001:** Overview of the impact of recreational drugs on the risk of hepatocellular carcinoma.

Drugs		Carcinogenic Potential Related to HCC	Physiopathology	Literature Support	References
Conventional drugs (ebrotidine, amiodarone, methotrexate and nitrofurantoin)		Presumed ^1^	Cirrhosis (cirrhosis–HCC sequence) Amiodarone: protective factor	No evidence Amiodarone data from retrospective observational study	[18,19,20,21,22,23,24] [25,26]
Herbal & Dietary Supplements	Khat Kava Pyrrolizidine alkaloids	Presumed Moderate ^2^ Presumed	Cirrhosis (cirrhosis–HCC sequence) Unknown Veno-occlusive syndrome (cirrhosis–HCC sequence)	No evidence Evidence in vivo, animal (case series) studies. None in humans No evidence	[27,28,29,30,31,32,33,34,35,36,37,38] [39,40,41,42,43,44,45,46,47,48,49,50,51,52,53] [54,55]
Tobacco		Strong ^3^	Carcinogenic substances after hepatic metabolism (via CYP2E1)	Evidence from retrospective/prospective observational studies and meta-analysis	[56,57,58,59]
Cannabis		Controversial ^4^	Fast progression of liver fibrosis in chronic C hepatitis	Evidence for: retrospective observational study; evidence against: prospective observational studies and meta-analysis	[60] [61,62,63]
Cocaine		Absent ^5^	NA	No evidence	NA
Heroin		Absent	NA	No evidence	NA
Amphetamines		Absent	NA	No evidence	NA
AAS		Strong	Genetic predisposition (?) Unknown liver-adenoma-like underlying disease (?)	Evidence based on cases and case series	[64,65,66,67,68,69,70,71,72,73,74,75,76,77,78,79,80,81,82,83]

NA: not applicable; 1: the agents are known to induce cirrhosis and, hence, might also promote HCC, 2: evidence on carcinogenic effects in animal models but lack of data on humans, 3: data enough to support a causal relationship between the exposure to the agent and appearance of HCC, 4: controversial data on the potential to accelerate fibrosis and, afterward, cirrhosis and HCC, 5: lack of data that allow researchers to link the agent with HCC.

**Table 2 cancers-14-05395-t002:** Cases of anabolic-androgenic-steroid-related liver tumors reported in the literature.

Authors	AAS	Duration of Treatment *	Indication	Single or Multiple Tumor	Initial Symptoms/ Signs	α-Fetoprotein (ng/mL)	Other Histological Findings
Bernstein et al. [64]	Oxymetholone	11	Fanconi´s anemia	Multiple	Yes	NA	Peliosis hepatis (tumoral and non-tumoral tissues)
Johnson et al. [65] (4 cases)	Oxymetholone Oxymetholone Methyltestosterone Methandienone	36 15 51 NA	Aplastic anemia Aplastic anemia Fanconi´s anemia Fanconi´s anemia	Multiple Multiple Multiple Single	Yes Yes Yes Yes	Not increased Not increased NA Not increased	NA NA NA NA
Henderson et al. [66]	Methyltestosterone Norethandrolone Stanozolol Oxymetholone **	90 **	Hypoplastic anemia	Multiple	Yes	High *	No
Farrell GC et al. [67] (3 cases)	Oxymetholone Methyltestosterone Methyltestosterone Testosterone **	65 72 96 **	PNH Hypopituitarism Cryptorchidism	Multiple Multiple Multiple	Yes Yes Yes	Not increased Not increased Not increased	NA NA NA
Hernández et al. [68]	Methandienone	36	PNH	Multiple	Yes	Not increased	Peliosis hepatis (in non-tumoral tissue)
Shapiro et al. [69]	Testosterone propionate Oxymetholone **	108 **	Fanconi´s anemia	Multiple	Yes	Not increased	Cholestasis, peliosis hepatis (in tumoral and non-tumoral tissues)
Lopez et al. [70]	NA	8	Aplastic anemia	Multiple	Yes	NA	NA
Carrasco et al. [71]	Nandrolone Testosterone enanthate **	132 **	Alport´s syndrome	Multiple	No	Not increased	Tumor cells in pseudo-acinar pattern. No peliosis hepatis
Linares et al. [72]	Oxymetholone	120	Fanconi´s anemia	Multiple	Yes	Not increased	Peliosis hepatis (in tumoral tissue)
Bork et al. [73] (3 cases)	Danazol Danazol Danazol	240 156 192	Hereditary angioedema Hereditary angioedema Hereditary angioedema	Single Multiple Single	No Yes No	NA NA NA	NA NA NA
Socas et al. [74] (2 cases)	Stanozolol Oxymetholone Nandrolone Testosterone Methenolone ** Stanozolol Oxymetholone Nandrolone Testosterone Boldenone **	180 ** 6 **	Recreational (bodybuilding) Recreational (bodybuilding)	Multiple Multiple	Yes Yes	Not increased Not increased	NA NA
Martin et al. [75]	Androstendione Nandrolone **	60 **	Recreational (bodybuilding)	Multiple	Yes	NA	Peliosis hepatis (in tumoral tissue)
Hardt et al. [76]	Testosterone Trenbolone acetate Androstanediol Boldenone Methandriol Letrozole Oxymetholone Methandienone **	60 **	Recreational (bodybuilding)	Single	Yes	Not increased	IHC: cytoplasmic CK8 (+), Hep-Par1 (+). Canalicular CD10 (+), CEA (+). Nuclear β-catenin, progesteron and estrogen receptors (weakly +).
Pais-Costa et al. [77]	Andronstendiona Nandrolone **	72 **	Recreational (bodybuilding)	Multiple	Yes	Not increased	NA
Kesler et al. [78]	Testosterone	84	Recreational (bodybuilding)	Multiple	Yes	High (366)	IHC: arginase, glypican 3, heat shock protein 70, glutamine synthetase, β-catenin and CD34 (+)
Solbach et al. [79]	Nandrolone Sustanon Methandienone Stanozolol **	72 **	Recreational (bodybuilding)	Multiple	Yes	NA	IHC: glutamine synthetase, androgen-receptor nuclear, β-catenin and CD34 (+)
Kato K et al. [80]	Testosterone enanthate	144	F-to-M gender identity disorder	Multiple	Yes	Not increased	IHC: β-catenin and GS (+)
Woodward et al. [81] (2 cases)	NA NA	NA 60	Recreational (bodybuilding) Recreational (bodybuilding)	Single Multiple	Yes Yes	NA NA	NA NA
Wang et al. [82]	Stanozolol	48	Aplastic anemia	Multiple	Yes	Not increased	IHC: β-catenin and CD34 (+)
Lin et al. [83]	Testosterone cypionate	14	F-to-M gender identity disorder	Multiple	Yes	High (4320)	IHC: androgen-receptor (+)

F-to-M: female to male; IHC: immunohistochemistry; NA: not available; PNH: paroxysmal nocturnal hemoglobinuria. * Time in months from the beginning of the first/unique treatment until liver tumor diagnosis. ** Multiple and stepwise AAS treatments during case evolution.

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
