# Peer review of "Recreational Drugs and the Risk of Hepatocellular Carcinoma"

_cancers, 2022, doi:10.3390/cancers14215395_

Round 1

Reviewer 1 Report

It is well known that cirrhosis is the main risk to develop hepatocarcinoma. Several factors such as viral hepatitis, alcohol consumption and metabolic disorders are related with the development of cirrhosis. However, the risk of illegal drugs consumption in chronic liver damage need to be explore.  The aim of this review is to provide updated information and to unify the available published evidence on legal and illegal recreational drug use and possibility to develop HCC.

This manuscript is already accepted to be published, from my point of view. This is the first review that summarize the evidence on recreational drugs use and risk ok HCC. However, there are some minor issues that could be addressed:

a)       Table 1 recapitulate the impact of recreational drugs on the risk of HCC. It would be better to include a description of each classification (suspected, moderate or strong) and references.

b)      A summary of conventional drugs should be included in table 1 (ebrotidine, amiodarone, methotrexate and nitrofurantoin).

c)       Mostly of the drugs are related with liver damage (fibrosis) but not directly with HCC. It would be nice to include information about which pathways are activated in the liver to promote liver damage a consequently HCC development.

d)      Due to mostly of these kinds of drugs are considered as Livertoxic (category A), it would be nice to include a column with this information or a paragraph with the description of each stage.

1)      Please, the first time you use an abbreviation, it’s important to spell out the full term and put the abbreviation in parentheses. In contrast, if abbreviations are not going to be use more than once time it not to be added. Please change them through the text.

Author Response

  1. Table 1 recapitulate the impact of recreational drugs on the risk of HCC. It would be better to include a description of each classification (suspected, moderate or strong) and references.

We have included a footnote to describe the meaning of each classification in Table 1, pag 3. Besides, we have included references in Table 1, pag 3.

  1. A summary of conventional drugs should be included in table 1 (ebrotidine, amiodarone, methotrexate and nitrofurantoin).

This information has been included in table 1, page 3

  1. Mostly of the drugs are related with liver damage (fibrosis) but not directly with HCC. It would be nice to include information about which pathways are activated in the liver to promote liver damage a consequently HCC development.

We thank the reviewer for her/his comments. Liver damage associated with drugs/xenobiotics involves several pathways but there is a general notion that oxidative stress plays a key role in inflammation and subsequently liver fibrosis. However, the activation of oncogenic pathways in this process is largely unknown. Hence, we think that including some sentences on that in the manuscript would not add value since it would be fully speculative.

  1. Due to mostly of these kinds of drugs are considered as Livertoxic (category A), it would be nice to include a column with this information or a paragraph with the description of each stage.

Good point. We have added the following sentences: “(this xenobiotic has been reported and known or highly likely to cause idiosiyncratic liver damage, between 12-50 cases have been previously published)” in pag 5, line 153-155, and (the xenobiotic is well known, with more than 50 cases published in the literature)” in pag 5, line 180-181, to explain the specific categorization of the likelihood of Drug Induced Liver Injury by Livertox.

  1. Please, the first time you use an abbreviation, it’s important to spell out the full term and put the abbreviation in parentheses. In contrast, if abbreviations are not going to be use more than once time it not to be added. Please change them through the text.

Done

Reviewer 2 Report

The authors provide a review on the risk of liver cancer after chronic use of recreational drugs (Khat, Kava, Tobacco, cannabis, cocaine, heroin, amphetamines, and anabolic steroids). The review is generally well-written and succinct, briefly touching on the potential risk of these recreational drugs to fibrosis and HCC. The review makes it apparent that there is a lack of research into this area for some drugs (cocaine, heroin, amphetamines) and the available evidence for the majority of these drugs suggests that their carcinogenic potential is limited (excluding anabolic steroids). The tables are a valuable addition. This review is suitable for publication but requires some modifications:

1.      Table 1 should provide citations, for example, the literature support column should provide references to the claims in the table, similar to what is done in Tables 2 and 3.

2.      Overall, throughout the review it is mainly a presentation of the results of previous studies without any additional synthesize or commentary explaining the contradictions between studies. For example, page 5 lines 176-180, mice exposed to kava developed neoplastic lesions but in human studies there is no evidence linking kava intake to HCC. This seems like an opportunity for the authors to add expert insight into why their may be a discrepancy between the rodent studies and humans. For example, is it a limitation of the mouse work (e.g. high doses not recapitulated in the human population?) or is it a limitation of the quality of the human studies (underpowered, short term exposure that requires more longitudinal investigations?). Every section would benefit by adding this sort of commentary. The section on anabolic steroids and HCC is the only section that provides this sort of critical review of the available literature.

3.      The cigarette smoking and HCC section: a summary of the key findings after line 220 (pg 6), that synthesizes the findings of the observational case-control and cohort studies would greatly improve the value of this section.

Minor comments:

1.      Pg 2 line 47, “considerable” should be “considerably”.

2.      Pg 2 line 79 “licit” should be “illicit”.

3.      Pg 4 line 142 “manly” should be “mainly”.

4.      Pg 4 line 144 “sold” should be “sale”.

Author Response

  1. Table 1 should provide citations, for example, the literature support column should provide references to the claims in the table, similar to what is done in Tables 2 and 3.

Done. We have included references in Table 1, pag 3.

  1. Overall, throughout the review it is mainly a presentation of the results of previous studies without any additional synthesize or commentary explaining the contradictions between studies. For example, page 5 lines 176-180, mice exposed to kava developed neoplastic lesions but in human studies there is no evidence linking kava intake to HCC. This seems like an opportunity for the authors to add expert insight into why their may be a discrepancy between the rodent studies and humans. For example, is it a limitation of the mouse work (e.g. high doses not recapitulated in the human population?) or is it a limitation of the quality of the human studies (underpowered, short term exposure that requires more longitudinal investigations?). Every section would benefit by adding this sort of commentary. The section on anabolic steroids and HCC is the only section that provides this sort of critical review of the available literature.

We thank the reviewer for his/her thoughtful comments. We have added the following sentence “This apparent discrepancy could be related to the low quality of the design of these studies (observational, lacking long-term follow-up) that precluding drawn firm conclusion” in pag 5-6, line 86-88.

  1. The cigarette smoking and HCC section: a summary of the key findings after line 220 (pg 6), that synthesizes the findings of the observational case-control and cohort studies would greatly improve the value of this section.

Good point, we have added the following sentence “In summary, tobacco smoking is a strong risk factor for HCC either alone or in association with other carcinogenic agents such as alcohol and probably potentiates other pre-cancer settings such as MAFLD” in pag 6, line 227-229.

Minor comments:

  1. Pg 2 line 47, “considerable” should be “considerably”.
  2. Pg 2 line 79 “licit” should be “illicit”.
  3. Pg 4 line 142 “manly” should be “mainly”.
  4. Pg 4 line 144 “sold” should be “sale”.

Thanks. All of them have been changed following your adjustments.
